# Outcomes of patients with malignant duodenal obstruction after receiving self-expandable metallic stents: A single center experience

Tien-Hsin Wei[1,2], Bing-Wei Ye[2,3], Pei-Shan Wu[2,4,5]*, Chung-Pin Li[2,4,6], Yee Chao[2,7], Pei-Chang Lee[2,4], Yi-Hsiang Huang[2,4], Kuei-Chuan Lee[2,4]*, Ming-Chih Hou[2,4,5]

1 Division of Gastroenterology and Hepatology, Department of Medicine, Taipei Municipal Gandau Hospital, Taipei, Taiwan, 2 Department of Medicine, School of Medicine, National Yang Ming Chiao Tung University, Taipei, Taiwan, 3 Division of Gastroenterology, Department of Medicine, Taiwan Adventist Hospital, Taipei, Taiwan, 4 Division of Gastroenterology and Hepatology, Department of Medicine, Taipei Veterans General Hospital, Taipei, Taiwan, 5 Endoscopy Center for Diagnosis and Treatment, Taipei Veterans General Hospital, Taipei, Taiwan, 6 Division of Clinical Skills Training, Department of Medical Education, Taipei Veterans General Hospital, Taipei, Taiwan, 7 Department of Oncology, Taipei Veterans General Hospital, Taipei, Taiwan

* kclee2@vghtpe.gov.tw (KCL); pswu6@vghtpe.gov.tw (PSW)

**Data Availability Statement:** All relevant data are within the article and its Supporting information files.

## Abstract

### Objectives

Self-expandable metallic stent (SEMS) placement is a safe and effective palliative treatment for malignant gastric outlet obstruction; however, the clinical outcomes of gastric and duodenal stenoses may differ. This study aimed to investigate the clinical efficacy of SEMS placement and the predictors of clinical outcomes, specifically in malignant duodenal obstruction (MDO).

### Methods

Between September 2009 and March 2021, 79 patients with MDO who received SEMS placement in our hospital were retrospectively enrolled. Patients were divided into three groups according to the obstruction levels: above-papilla group (type 1), papilla involved group (type 2), and below-papilla group (type 3). The clinical outcomes and predictors of survival and restenosis were analyzed.

### Results

The technical and clinical success rates were 97.5% and 80.5%, respectively. Among patients who had successful stent placement, stent restenosis occurred in 17 patients (22.1%). The overall median stent patency time was 103 days. The overall median survival time after stent placement was 116 days. There was no difference in the stent patency, or stent dysfunction and procedure-related adverse events among the three groups. A longer length of duodenal stenosis ≥ 4 cm was associated with poor prognosis (hazard ratio [HR] =

**Funding:** This study was in part supported by the Taipei Veterans General Hospital (Grant No. V110A-001 and V110C-143) and the Ministry of Science and Technology (Grant No. MOST 110-2628-B-075 -016).

**Competing interests:** The authors have declared that no competing interests exist.

1.92, 95% confidence interval [CI] = 1.06–3.49, $p$ = 0.032) and post-stent chemotherapy was associated with lower mortality (HR = 0.33; 95% CI = 0.17–0.63, $p$ = 0.001).

## Conclusion

SEMS is a safe and effective treatment for MDO. Chemotherapy after SEMS implantation improve the survival for these patients and a longer length of stenosis predicts higher mortality.

## Introduction

Malignant gastroduodenal obstruction is a troublesome late complication of gastrointestinal, pancreatobiliary malignancies, and cancers with gastroduodenal metastases. Persistent nausea, vomiting, and poor appetite develop in these patients, which lead to a rapid decline in the nutritional status and deterioration in their quality of life [1–5]. Palliative chemotherapy and/ or radiotherapy are difficult to administer to these patients. Therefore, the prognosis of patients with malignant gastroduodenal obstructions is poor, with a median survival of three to four months [6].

Patients with malignant gastroduodenal obstruction have traditionally been treated with surgical bypass for palliative management. Although gastrojejunostomy has a good clinical outcome, some patients might not be suitable for surgical intervention because of the unstable clinical conditions. Therefore, self-expandable metallic stent (SEMS) placement has been considered as an alternative palliative intervention recently [7, 8]. The advantages of SEMS over gastrojejunostomy include immediate symptom improvement, earlier resumption of oral intake, shorter length of hospital stays, less cost, lower morbidity, and mortality rates [9–12]. Studies have shown impressive results of SEMS placement in gastric outlet obstruction. In a systematic review, Jeurnink *et al.* reported high technical success rate of 96% and clinical success rate of 89% in patients with gastric outlet obstruction and underwent SEMS placement [8]. Nevertheless, the clinical outcomes might be different between patients with gastric stenosis and duodenal stenosis after SEMS placement. In fact, stent placement beyond the duodenal bulb is more technically challenging given the complexity of managing strictures involving the major papilla and/or the common bile duct. Further, endoscopists may encounter loop formation of the endoscopes in the stomach or the tortuous duodenum, which leads to the difficult in deploying stents. However, limited studies have discussed specifically on the outcomes of malignant duodenal obstruction (MDO) with SEMS placement. Chiu *et al.* reported that in patients with obstruction level beyond the duodenal bulb, the technical success rate of was 87% and clinical success rate was 93% for stent placement [13]. Wu *et al.* further compared the clinical outcomes between patients who received SEMS for the obstruction level above the major papilla (including papilla itself) and below the major papilla, with technical success rates of 100% and 93%, respectively. The clinical success rates were 100% in both the above-papilla and below-papilla groups [14]. Though these studies disclosed that palliative stenting in duodenal obstruction is clinically effective, factors that influence the prognosis in these patients remain unclear.

In this study, we aimed to assess the efficacy and safety of duodenal SEMS placement and investigate the predictors of stent patency and post-stent survival.

## Method

### Patients

We retrospectively reviewed the medical records of 79 consecutive patients with inoperable MDO who underwent duodenal SEMS placement at Taipei Veterans General Hospital between September 2009 and March 2021. The diagnosis of duodenal obstruction was based on endoscopic and/or imaging findings. All patients were followed up until August 2021. Patients who had previously underwent surgical gastroenterostomy were excluded from this study. This study was approved by the Institutional Review Board of Taipei Veteran's General Hospital (IRB No. 2020-06-035BC).

### Stent placement

Computed tomography and upper gastrointestinal endoscopy were performed to evaluate the site, severity, and length of stenosis before stent placement. All patients received nothing orally and received nasogastric tube drainage to relieve symptoms of obstruction and to minimize the risk of aspiration during the procedure. The patients were treated in the supine position and the severity of the duodenal stenosis was evaluated by the endoscope with local anesthesia. Under fluoroscopic guidance, the SEMS was deployed using an upper gastrointestinal endoscope (GIF-2T240 or GIF-2TQ260M; Olympus, Tokyo, Japan), duodenoscope (JF-260) or colonoscope (HQ-290I), depending on the sites of obstruction. As the endoscope was advanced to the stenotic site, a guide wire (Hydra Jagwire, Boston Scientific Corporation, USA) was introduced into the working channel of the scope more than 20 cm beyond the obstruction level. Then, the location and length of the stenosis were assessed by injecting a water-soluble contrast medium. The length of the stent was determined according to the lesion length, with an additional 2–3 cm on each end at stenotic site. Uncovered stents such as the WallFlex Single-use Duodenal Stent (Boston Scientific Corporation, USA) or Bonastent (Standard SCI. Tech Inc, South Korea) were deployed under fluoroscopic guidance, as in our previously published studies [15, 16]. An abdominal plain film was performed the day after the procedure to ensure the expansion and location of the SEMS. The patient resumed oral intake after confirming the stent's condition. Once patients developed recurrent obstructive symptoms, further imaging studies, such as computed tomography or endoscopy, would be arranged. Another stent would be inserted if restenosis was present.

### Data collection

All data were obtained from the patients' medical records, including procedure notes, radiology, and endoscopy reports. The patients were divided into three groups. The above-papilla group (type 1) was defined as tumor obstruction at the level above the papilla and does not involve the papilla; the papilla involved group (type 2) was defined as tumor affecting the second portion of the duodenum, with papilla involvement; the below-papilla group (type 3) had duodenal obstruction distal to the papilla and does not involve the papilla [17]. The collected data included patient demographics and procedure characteristics. The outcomes that were measured included the technical and clinical success rates, gastric outlet obstruction scoring system (GOOSS) scores before and after the procedure, procedure-related adverse events, duration of stent patency, and survival time.

Technical success was defined as successful deployment of the SEMS across the stenotic site. Clinical success was defined as symptomatic relief and was measured by improvement of the GOOSS score [16, 18]. The GOOSS score was measured by evaluating the status of diet intake, in which 0 = no oral intake, 1 = liquid diet only, 2 = soft solids, and 3 = low-residue or

full diet [19]. The details for the diets are shown in S1 Table. Procedure-related adverse events were classified as major and minor events. Major events included life-threatening or severe complications, such as aspiration pneumonia, perforation, sepsis, or bleeding. Minor events were defined as complications that were not life-threatening and did not require aggressive management, including abdominal pain, nausea, and vomiting.

## Statistical analysis

Continuous variables are presented as mean ± standard deviation or median and range. Categorical variables are presented as frequencies and percentages. Differences in the data between the above-papilla (type 1), papilla involved (type 2), and the below-papilla (type 3) groups were compared using the chi-squared test or Fisher's exact test for categorical variables, and the Kruskal-Wallis test for continuous variables. Univariate and multivariate Cox regression analysis were performed to identify predictors for clinical outcomes. Only variables with $p < 0.1$ in univariate analysis were included in multivariate Cox regression models. Univariate analysis of survival time and stent patency were performed using Kaplan-Meier analysis. Statistical significance was defined as a two-sided $p$ value $< 0.05$. All statistical analyses were performed using SPSS Statistics for Windows, version 24.0 (IBM, Armonk, NY, USA).

## Results

### Patient characteristics

A total of 79 patients with MDO were enrolled and baseline characteristics for these patients are presented in Table 1. Of the 79 patients, 44 (55.7%) were male and 35 (44.3%) were female, with a mean age of 65.8 years. All the patients with MDO were in stage III or IV. Pancreatic cancer was the main cause of duodenal obstruction (78.5%), followed by duodenal cancer (7.6%). Most patients (79.7%) had significant weight loss (range: 8.0 ± 4.7 kg) within 6 months before the procedure and 26.6% of the patients had a poor performance status (ECOG score $\geq$ 3). Type 2 group is the most common type of obstruction (43.0%). More patients (88.2%, $p < 0.001$) in type 2 group had received biliary drainage as compared to those in the other two groups.

### Clinical outcomes

The clinical outcomes are summarized in Table 2. The technical and clinical success rate was 97.5% and 80.5%, respectively. The procedure time did not differ among the groups ($p = 0.354$), although the type 2 group tended to have a longer procedure time than the other two groups. The mean GOOSS scores improved from 0.27 before stent placement (day 0) to 1.92 on day 7 ($p < 0.001$), and to 2.23 on day 30 ($p < 0.001$) after the procedure (S1 Fig). Stent dysfunction was observed in 18 patients (23.3%), including 17 patients with restenosis and one patient with stent fracture. Among the restenosis events, 16 events were caused by tumor ingrowth, and one was caused by out-growth. Seventeen patients, including one patient with stent fracture, underwent reintervention for stent dysfunction. Fifteen patients received a second stent, and one of whom received a third stent to maintain lumen patency. One patient underwent jejuno-jejunal bypass, and one patient received a feeding jejunostomy. The only patient who did not receive reintervention had bowel perforation due to tumor invasion and refused further treatment due to the terminal condition. The major procedure related adverse event rate was 6.3%, with hemorrhage being the most common event. All three bleeding events were caused by tumor-related bleeding and only one patient required endoscopic hemostasis. None of the patients died due to major adverse events. Additionally, 34.2% of patients

**Table 1. Patient characteristics.**

| | Total (n = 79) | Type 1 (n = 17) | Type 2 (n = 34) | Type 3 (n = 28) | p-value |
|---|---|---|---|---|---|
| **Male gender** | 44 (55.7) | 8 (47.1) | 18 (52.9) | 18 (64.3) | 0.514 |
| **Age, years** | 65.8 ± 13.9 | 63.0 ± 10.8 | 68.6 ± 12.8 | 64.3 ± 16.4 | 0.282 |
| **Weight loss, n (%)** | 63 (79.7) | 14 (82.4) | 26 (76.5) | 23 (82.1) | 0.878 |
| **Baseline performance status (ECOG score), n (%)** | | | | | |
| 0 | 1 (1.3) | 1 (5.9) | 0 (0) | 0 (0) | |
| 1 | 29 (36.7) | 9 (52.9) | 9 (26.5) | 11 (39.3) | |
| 2 | 28 (35.4) | 5 (29.4) | 14 (41.2) | 9 (32.1) | |
| 3 | 17 (21.5) | 1 (5.9) | 11 (32.4) | 5 (17.9) | |
| 4 | 4 (5.1) | 1 (5.9) | 0 (0) | 3 (10.7) | |
| ≥ 3 | 21 (26.6) | 2 (11.8) | 11 (32.4) | 8 (28.6) | 0.280 |
| **Primary malignancy, n (%)** | | | | | |
| Duodenal cancer | 6 (7.6) | 1 (5.9) | 2 (5.9) | 3 (10.7) | |
| Pancreatic cancer | 62 (78.5) | 13 (76.5) | 25 (73.5) | 24 (85.7) | 0.497 |
| Cholangiocarcinoma | 4 (5.1) | 3 (17.6) | 1 (2.9) | 0 (0) | |
| Others † | 7 (8.9) | 0 (0) | 6 (17.6) | 1 (3.6) | |
| **Tumor stage, n (%)** | | | | | |
| III | 14 (17.7) | 2 (11.8) | 5 (14.7) | 7 (25.0) | 0.551 |
| IV | 65 (82.3) | 15 (88.2) | 29 (85.3) | 21 (75.0) | 0.551 |
| **Peritoneal carcinomatosis, n (%)** | 31 (39.2) | 11 (64.7) | 10 (29.4) | 10 (35.7) | 0.054 |
| **Liver metastasis, n (%)** | 36 (45.6) | 7 (41.2) | 20 (58.8) | 9 (32.1) | 0.110 |
| **Prior RT, n (%)** | 15 (19.0) | 6 (35.3) | 6 (17.6) | 3 (10.7) | 0.143 |
| **Post-stent RT, n (%)** | 9 (11.4) | 1 (5.9) | 4 (11.8) | 4 (14.3) | 0.819 |
| **Prior C/T, n (%)** | 40 (50.6) | 13 (76.5) | 15 (44.1) | 12 (42.9) | 0.059 |
| **Post-stent C/T, n (%)** | 44 (55.7) | 12 (70.6) | 13 (38.2) | 19 (67.9) | 0.026 |
| **Biliary drainage, n (%)** | 48 (60.8) | 10 (58.8) | 30 (88.2) | 8 (28.6) | <0.001 |

Data are expressed as number (%), mean ± standard deviation or median (range)

ECOG, Eastern Cooperative Oncology Group; RT, radiotherapy; C/T, chemotherapy

†Others include three gallbladder cancer, 2 retroperitoneum sarcoma, 1 colon neuroendocrine tumor, and 1 breast cancer.

developed minor adverse events and abdominal pain was the main complaint. No stent migration or duodenal perforation was observed and none of the patient died due to procedure-related complications in our study.

While comparing the clinical outcomes among the three subgroups, there was no statistical difference in the technical success rate, clinical success rate or procedure-related adverse events. The overall median stent patency time of the first stent was 103 days (range, 7–671 days) and there were no statistically significant differences among the three groups (p = 0.353). The median survival time after stent placement was 116 days (range, 10–887 days), with a significantly lower median survival time in the type 2 group (p = 0.024, Table 2, S2 Fig).

When comparing the clinical outcomes between patients with pancreatic cancer or non-pancreatic cancers, there was no statistic difference in survival time [121 (11–713) versus 78 (10–887) days, p = 0.737] and stent patency time [116 (10–671) versus 70 (7–564) days, p = 0.854]. Twenty-one patients received WallFlex stents and 56 patients received Bonastents. There was no significant difference in terms of survival time [120.0 (11–541) versus 118.5 (10–887) days, p = 0.455] and stent patency time [106.0 (11–541) versus. 106.0 (7–671) days, p = 0.544] between patients receiving WallFlex stents or Bonastents.

### Predictive factors for stent restenosis and survival

Univariate analysis for restenosis revealed no significant predictive factors for stent restenosis (S2 Table). In the univariate analysis for mortality, advanced performance status with ECOG score $\geq 3$, stenosis type with papilla involvement, longer length of duodenal stenosis ($\geq 4$ cm), and presence of liver metastasis were predictors of mortality. Post-procedure chemotherapy as well as biliary drainage prior to the procedure were associated with better survival. On multivariate analysis, duodenal stenosis length $\geq 4$ cm could predict mortality (HR = 1.92, 95% CI = 1.06–3.49, p = 0.032) (Table 3, Fig 1A). In addition, post-procedure chemotherapy was a predictor of lower mortality (HR = 0.33; 95% CI = 0.17–0.63, $p$ = 0.001). (Table 3, Fig 1B). We further analyzed the median survival of patients with pancreatic and non-pancreatic cancers (S3 Table). Patients with pancreatic cancer had a longer median survival time after receiving post-stent chemotherapy than those who did not (161 [21–713] versus 65 [11–268] days, $p$ <0.001). However, the median survival time did not differ between non-pancreatic cancer patients who were treated with versus without post-stent chemotherapy (142 [26–887] versus 71 [10–470] days, $p$ = 0.336).

## Discussion

In the present study, the technical and clinical success rates of SEMS placement were high in patients with MDO. Although the stent patency time was similar among type 1 to 3 duodenal

**Table 2. Clinical outcomes after self-expandable metallic stent placement.**

| | Total (n = 79)[†] | Type 1 (n = 17) | Type 2 (n = 34) | Type 3 (n = 28) | *p*-value |
|---|---|---|---|---|---|
| **Technical success, *n* (%)** | 77 (97.5) | 17 (100) | 33 (97.1) | 27 (96.4) | 1.000 |
| **Clinical success, *n* (%)** | 62 (80.5) | 14 (82.4) | 24 (72.7) | 24 (88.9) | 0.327 |
| **Procedure time, minute** | 26 ± 15 | 20 ± 3 | 28 ± 18 | 27 ± 13 | 0.354 |
| **Stent patency, day** | 103 (7–671) | 125 (26–551) | 70 (10–671) | 122 (7–564) | 0.353 |
| **Post-stent survival, day** | 116 (10–887) | 135 (26–713) | 78 (10–671) | 126 (21–887) | 0.024 |
| **Stent dysfunction, *n* (%)** | | | | | |
| Restenosis | 17 (22.1) | 3 (17.6) | 8 (23.5) | 6 (22.2) | 0.939 |
| Migration | 0 (0) | 0 (0) | 0 (0) | 0 (0) | n/a |
| Fracture | 1 (1.3) | 0 (0) | 1 (3.0) | 0 (0) | 1.000 |
| **Reintervention, *n* (%)** | 17 (22.1) | 3 (17.6) | 8 (23.5) | 6 (22.2) | 0.868 |
| **Procedure-related adverse events** | | | | | |
| **Major adverse event, *n* (%)** | 5 (6.3) | 2 (11.8) | 2 (5.9) [‡] | 1 (3.6) | 0.613 |
| Aspiration pneumonia | 1 (1.3) | 0 (0) | 1 (2.9) | 0 (0) | 1.000 |
| Perforation | 0 (0) | 0 (0) | 0 (0) | 0 (0) | n/a |
| Sepsis | 2 (2.5) | 0 (0) | 1 (2.9) | 1 (3.6) | 1.000 |
| Hemorrhage | 3 (3.8) | 2 (11.8) | 1 (2.9) | 0 (0) | 0.157 |
| **Minor adverse event, *n* (%)** | 27 (34.2) [§] | 4 (23.5) | 13 (38.2) | 10 (35.7) | 0.616 |
| Abdominal pain | 22 (27.8) | 3 (17.7) | 12 (35.3) | 7 (25.0) | 0.423 |
| Nausea | 13 (16.5) | 2 (11.8) | 5 (14.7) | 6 (21.4) | 0.734 |
| Vomiting | 14 (17.7) | 2 (11.8) | 6 (17.6) | 6 (21.4) | 0.752 |

Data are expressed as number (%), mean ± standard deviation, or median (range).

[†] Technical success and clinical success are analyzed for 79 patients and other variables are analyzed for 77 patients after excluding patients failed to receive stent placement.

[‡] One patient suffered from both aspiration pneumonia and gastrointestinal hemorrhage

[§] A patient could experience more than one minor adverse event

n/a, not applicable

**Table 3. Univariate and multivariate analysis of mortality.**

| Variables | Univariate analysis | | | Multivariate analysis | | |
|---|---|---|---|---|---|---|
| | n | Death, n (%) | p-value | HR | 95% CI | p-value |
| Age (≥ 70/ < 70 years old) | 31/46 | 25/37 (80.6/80.4) | 0.269 | | | |
| Sex (male/female) | 43/34 | 35/27 (81.4/79.4) | 0.389 | | | |
| ECOG score (≥ 3/ < 3) | 20/57 | 19/43 (95.0/75.4) | 0.006 | 1.47 | 0.75–2.87 | 0.258 |
| Tumor origin (Pancreatic cancer/ non-pancreatic cancer) | 61/16 | 51/11 (83.6/68.8) | 0.568 | | | |
| Stage (IV/III) | 63/14 | 49/13 (77.8/92.9) | 0.957 | | | |
| Location of obstruction (Papilla not involved/ Papilla involved) | 44/33 | 32/30 (72.7/90.9) | 0.008 | 0.96 | 0.78–1.19 | 0.728 |
| Length of stenosis (≥ 4/ < 4 cm) | 46/31 | 41/21 (89.1/67.7) | 0.022 | 1.92 | 1.06–3.49 | 0.032 |
| Length of stent (≥ 9/ < 9 cm) | 56/21 | 46/16 (82.1/76.2) | 0.245 | | | |
| Prior RT (yes/no) | 14/63 | 12/50 (85.7/79.4) | 0.339 | | | |
| Post-stent RT (yes/no) | 9/68 | 8/54 (88.9/79.4) | 0.750 | | | |
| Prior C/T (yes/no) | 38/39 | 29/33 (76.3/84.6) | 0.588 | | | |
| Post-stent C/T (yes/no) | 44/33 | 33/29 (75.0/87.9) | <0.001 | 0.33 | 0.17–0.63 | 0.001 |
| Biliary drainage (yes/no) | 47/30 | 43/19 (91.5/63.3) | 0.021 | 1.61 | 0.89–2.93 | 0.116 |
| Peritoneal carcinomatosis (yes/no) | 30/47 | 22/40 (73.3/85.1) | 0.421 | | | |
| Liver metastasis (yes/no) | 34/43 | 29/33 (85.3/76.7) | 0.046 | 1.65 | 0.98–2.79 | 0.061 |

HR, hazard ratio; CI, confidence interval; RT, radiotherapy; C/T, chemotherapy; ECOG, Eastern Cooperative Oncology Group

obstruction types, patients with papilla involvement (type 2) had a shorter survival time than those without papilla involvement (type 1 and type 3). A longer length of stenosis was a predictor for higher mortality, whereas post-stent chemotherapy potentially helped to improve survival for pancreatic cancer with duodenal obstruction. These findings may help clinical physicians in deciding the best treatment strategies for patients with MDO.

With the advances in oncological treatments such as chemotherapy, targeted therapy, and immunotherapy in recent decades, patients with advanced gastrointestinal malignancies have improved survival [20–27]. Therefore, management of gastroduodenal obstruction to resume nutrition status and quality of life for these patients is important. Several studies have demonstrated the high efficacy and safety of endoscopic stenting for gastroduodenal obstruction [2, 6, 12, 28–30]. Nevertheless, there is a paucity of information about the clinical outcome of metallic stenting in MDO, especially the outcomes at different portions of the duodenum. Compared with pyloric or duodenal bulb obstruction, stenting below the duodenal bulb is much more technically challenging. Because of the distance of the stenosis site and the acute angle of the 2nd to 3rd portion or 3rd to 4th portion of the duodenum, it is difficult for an endoscopist to deliver the endoscope to the stenosis site and to release the stent in the angulated intestine. Despite these difficulties mentioned, Wu et al. disclosed an equal efficacy of stenting between the patients with duodenal obstruction levels above and below the major papilla, with no increased adverse events in the below papilla group [14]. However, the long-term outcomes following metallic stenting on different duodenal portions and predictors for prognosis were not analyzed in their study. To better understand the clinical outcomes of metallic stenting for duodenal obstruction at different locations, we divided patients into three obstruction levels: the above-papilla group (type 1), the papilla involved group (type 2), and the below-papilla group (type 3). The overall technical and clinical success rates were high in patients with duodenal obstruction, similar to the results in patients with gastroduodenal stenting [8, 14, 31]. Consistent with the findings of Wu et al., though technical difficult for stenting at distal duodenum, patients with type 2 and 3 obstruction had comparable technical/ clinical success rates compared with type 1

**(A)**

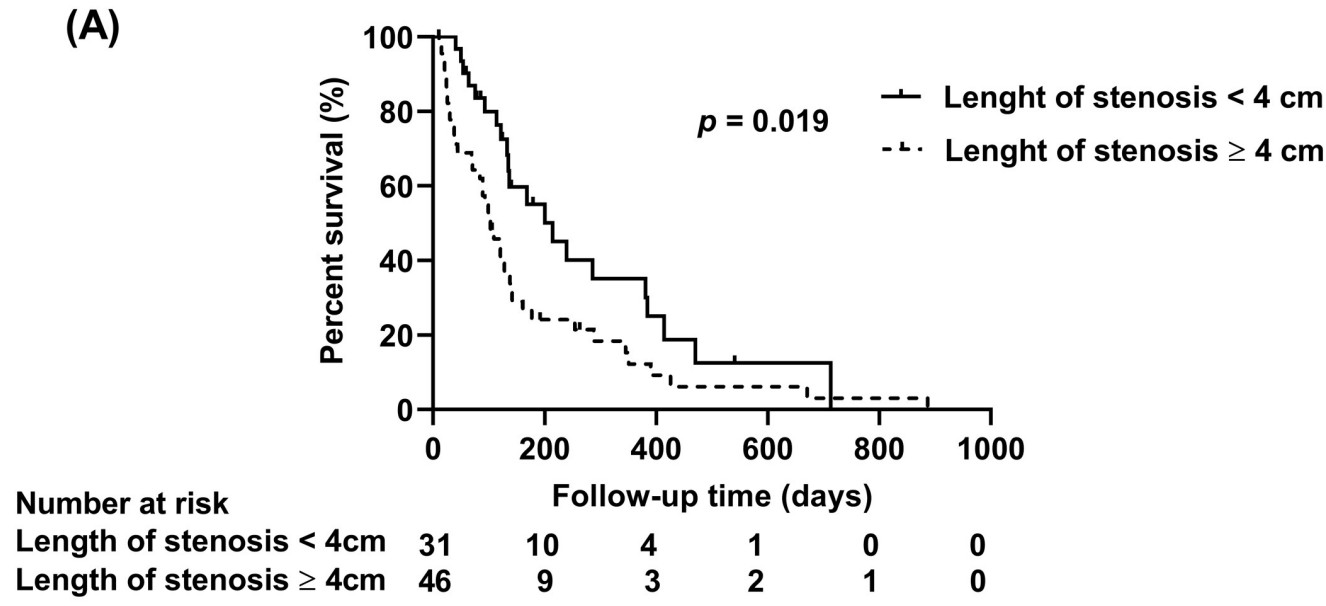

**(B)**

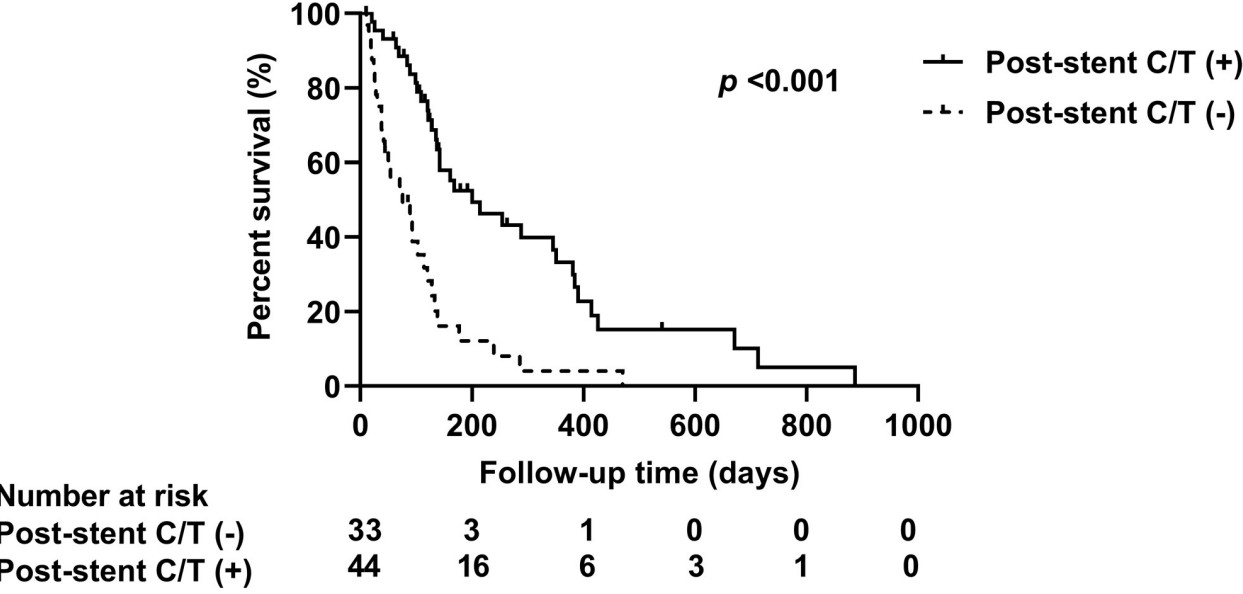

**Fig 1. Kaplan-Meier survival curves for post-stent survival in patients with malignant duodena obstruction.** (A) Length of stenosis < 4cm versus ≥ 4cm. (B) Post-stent C/T (-) versus post-stent C/T (+). C/T, chemotherapy. The *p*-value corresponds to log–rank test.

obstruction. There was no statistical significance about the procedure time among the three groups in our study. However, patients with type 2 and type 3 obstruction tended to have longer procedure time than those with type 1 obstruction (28 ± 18 and 27 ± 13 versus 20 ± 3 min, respectively). Similarly, a trend of longer procedure time was observed in patients with an

obstruction level below the papilla in Wu's study (23.1 ± 14.7 versus. 18.6 ± 11.1 minutes, $p$ = 0.123). In the present study, the comparison of the clinical outcomes among the three groups did not reveal significant differences in the stent patency time or procedure-related adverse events. Moreover, the type 2 group had a shorter median survival time with less post-stent chemotherapy than the other two groups. It may be attributed that more than 80% of patients with type 2 duodenal obstruction require biliary drainage for obstructive jaundice. Even after adequate biliary drainage, these patients might still experience recurrent biliary tract infections due to poor immune status, which would interrupt their cancer treatment. Chemotherapy plays an important role in cancer treatment for our patient population. Therefore, the interruption of chemotherapy may have contributed to the poor prognosis in the type 2 group.

Patients with gastric outlet obstruction had poor prognosis even after stent placement, with the post-stent survival time ranged from two to five months [8, 32–34]. Similarly, a short life expectancy was found for patients who had MDO, with a median survival approximately 3.9 months in our study. Oh *et al.* had reported that chemotherapy improved post-stent survival in malignant gastric outlet obstruction, both for patients with pancreatic cancer and non-pancreatic cancers [32]. Consistently, we also found that post-chemotherapy improved survival for patients with malignancy-related duodenal obstruction. Further analysis showed that the median survival was significantly longer in patients with pancreatic cancer treated with post-stent chemotherapy, which echoed our previously published study findings that post-stent chemotherapy prolonged survival in pancreatic cancer with gastric outlet obstruction [35]. However, for patients with non-pancreatic cancers, there was no significant survival benefit for patients who received post-stent chemotherapy. The long-term survival of patients with advanced pancreatic adenocarcinoma has been increasing over time as newer chemotherapy regimens, such as FOL-FIRINOX [25], gemcitabine/nab-paclitaxel [24], and nanoliposomal irinotecan in combination with 5-FU and leucovorin [23], have been applied to these patients. The role of post-stent chemotherapy in the survival of pancreatic cancer could be important. For the other tumor types, there were limited patient numbers and the large diversity of chemotherapy regimens as well as combination therapies, such as targeted therapy and immunotherapy, among cancers. Therefore, the effectiveness of chemotherapy in different cancers could not be properly analyzed in this study. Further studies with larger sample sizes are needed to validate the role of chemotherapy in different cancers with MDO. Oh *et al.* also mentioned that the effect of post-stent chemotherapy for gastric outlet obstruction might be attributed to selection bias regarding better clinical condition in the chemotherapy group [32]. In our study, although patients who received chemotherapy had better ECOG scores than those without chemotherapy, there was no statistical significance for ECOG scores on Cox regression analysis for survival. Additionally, we found that duodenal stenosis more than 4 cm is associated with a higher risk of post-stent mortality. A longer length of stenosis may be associated with a larger tumor burden, resulting in poor post stent prognosis. Briefly, once MDO develops, the nutritional status and performance status decline rapidly, resulting in interruption or inability for chemotherapy treatment. Endoscopic stent placement helps patients to relieve obstruction symptoms and resume nutrition. The data in our study provide information to encourage patients to continue or start chemotherapy once the obstruction symptoms are relieved, especially patients with pancreatic cancer.

This study has several limitations. First, as this was a retrospective study using data from a single medical center with a small sample size, caution must be taken when interpreting the data. Second, most of the MDO cases in our study was caused by pancreatic cancer, and the data could not be fully applied to patients with non-pancreatic cancer related MDO. Third, there is a great diversity of chemotherapy regimens among different cancers and most patients were enrolled before the era of immunotherapy. Therefore, further studies are needed to determine the optimal palliative treatment strategies for these patients after stent insertion.

In conclusion, endoscopic duodenal stenting is effective and safe for relieving symptoms of obstruction in patients with MDO. A longer length of stenosis predicted poorer clinical outcomes and post-stent chemotherapy may help to prolong the survival for these patients, especially in patients with pancreatic cancer.

## Supporting information

**S1 Table. Description of the patients' level of oral intake according to the GOOSS score.**
(DOCX)

**S2 Table. Univariate analysis of stent restenosis.**
(DOCX)

**S3 Table. Median survival in patients with and without post-stent chemotherapy.**
(DOCX)

**S1 Fig. Gastric Outlet Obstruction Score System (GOOSS) scores at Day 0, Day 1, Day 7, and Day 30 after metallic stent placement.** ns, not significant ****: $p < 0.0001$ between the groups determined by Kruskal Wallis test.
(TIF)

**S2 Fig. Kaplan-Meier survival curves for post-stent survival in patients with malignant duodena obstruction.** Type 1: the above-papilla group; type 2: the papilla involved group; type 3: the below-papilla group (type 3). The *p*-value corresponds to log–rank test.
(TIF)

## Acknowledgments

The authors gratefully acknowledge Mr. Dong-Ming Liao for his excellent technical assistance.

## Author Contributions

**Conceptualization:** Bing-Wei Ye, Kuei-Chuan Lee.

**Data curation:** Tien-Hsin Wei, Bing-Wei Ye, Chung-Pin Li, Yee Chao, Pei-Chang Lee, Yi-Hsiang Huang.

**Formal analysis:** Tien-Hsin Wei, Chung-Pin Li, Yee Chao, Pei-Chang Lee, Yi-Hsiang Huang.

**Funding acquisition:** Pei-Shan Wu.

**Methodology:** Tien-Hsin Wei, Bing-Wei Ye.

**Supervision:** Pei-Shan Wu, Kuei-Chuan Lee, Ming-Chih Hou.

**Validation:** Pei-Shan Wu, Kuei-Chuan Lee, Ming-Chih Hou.

**Visualization:** Pei-Shan Wu, Ming-Chih Hou.

**Writing – original draft:** Tien-Hsin Wei.

**Writing – review & editing:** Pei-Shan Wu, Kuei-Chuan Lee.

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
