## [Decision Letter · Decision Letter 0]

15 Feb 2022

PONE-D-22-02289Outcomes of Patients with Malignant Duodenal Obstruction After Receiving Self-Expandable Metallic Stents: A Single Center ExperiencePLOS ONE

Dear Dr. Kuei-Chuan Lee,

Thank you for submitting your manuscript to PLOS ONE. After careful consideration, we feel that it has merit but does not fully meet PLOS ONE’s publication criteria as it currently stands. Therefore, we invite you to submit a revised version of the manuscript that addresses the points raised during the review process.

We look forward to receiving your revised manuscript.

Kind regards,

Wenguo Cui, Ph.D

Academic Editor

PLOS ONE

Journal Requirements:

Reviewers' comments:

Reviewer's Responses to Questions

**Comments to the Author**

1. Is the manuscript technically sound, and do the data support the conclusions?

Reviewer #1: Partly

Reviewer #2: Yes

2. Has the statistical analysis been performed appropriately and rigorously? 

Reviewer #1: Yes

Reviewer #2: Yes

3. Have the authors made all data underlying the findings in their manuscript fully available?

Reviewer #1: Yes

Reviewer #2: Yes

4. Is the manuscript presented in an intelligible fashion and written in standard English?

Reviewer #1: Yes

Reviewer #2: Yes

5. Review Comments to the Author

Reviewer #1: This manuscript mainly reported a kind of self-expandable metallic stent (SEMS) placement in 79 patients with MDO between September 2009 and March 2021, who were divided into three groups according to the obstruction levels. The author claims that implantation improve the survival for these patients and a longer length of stenosis predicts higher mortality. However, still has some major issue which impairs the conclusion raised by authors.

1.In this paper, the stenosis is divided into types 1, 2, and 3 (above-papilla group (type 1), papilla involved group (type 2), and below-papilla group (type 3). In many cases, the stenosis caused by the tumor is also combined, as 1 and 2 combined, or 2 and 3 combined. This article does not consider or mention this special case.

2.There is bias in case selection. Different tumor types have different growth rates. After stent placement, the factors that lead to restenosis and shedding are also different. This manuscript did not clarify this part.

3.The authors only mentions the length of the stenosis, while no the degree of stenosis. The degree of stenosis in the same part of the stenosis is different, and the effect after placement is also different.

4.The type of stent placed was not specified. The same type of stent bears the same support force and different stents bear different support forces.

Reviewer #2: The work Outcomes of Patients with Malignant Duodenal Obstruction After Receiving Self-Expandable Metallic Stents: A Single Center Experience summarized data from 79 malignant duodenal obstruction patients treated with self-Expandable Metallic Stents. Study is properly designed and executed and I have few comments for the work.

First of all, author used ECOG score to indicate patients health condition. Table 1 and 3 should follow Table S1 change ECOG to ECOG score. Or the abbreviation of the Oncology Group will not give any meaning for the numbers.

Second, GOOSS score was summarized till day 30. Since patients did not receive chemotherapy after put in stent has a median survival around 60 days. Is there any GOOSS score data for patients till day 60, 90?

Last but not the least, GOOSS score largely influenced patient recovery due to the nutrient acquired always determined by diet. To understand the recovery progress, is it possible to include a GOOSS score chart with detailed classification.

6. PLOS authors have the option to publish the peer review history of their article (what does this mean?). If published, this will include your full peer review and any attached files.

Reviewer #1: No

Reviewer #2: No

---

## [Author Response · Author response to Decision Letter 0]

18 Feb 2022

Reviewer #1: This manuscript mainly reported a kind of self-expandable metallic stent (SEMS) placement in 79 patients with MDO between September 2009 and March 2021, who were divided into three groups according to the obstruction levels. The author claims that implantation improve the survival for these patients and a longer length of stenosis predicts higher mortality. However, still has some major issue which impairs the conclusion raised by authors.

1.In this paper, the stenosis is divided into types 1, 2, and 3 (above-papilla group (type 1), papilla involved group (type 2), and below-papilla group (type 3). In many cases, the stenosis caused by the tumor is also combined, as 1 and 2 combined, or 2 and 3 combined. This article does not consider or mention this special case.

Reply: Thank you for the comments. The above-papilla group (type 1) is defined as tumor obstruction at the level above the papilla and does not involve the papilla; the papilla involved group (type 2) is defined as tumors affecting the second portion of the duodenum and involve the papilla; the below-papilla group (type 3) has duodenal obstruction distal to the papilla and does not involve the papilla. Therefore, there is no combined case according to the definitions in our study. The detailed definitions of the three MDO types are added in the manuscript. Please page 9-10, line 155-159.

2.There is bias in case selection. Different tumor types have different growth rates. After stent placement, the factors that lead to restenosis and shedding are also different. This manuscript did not clarify this part.

Reply: Thank you for the important opinions. In the present study, pancreatic cancer is the major etiology of MDO (62 patients, 78.5%), followed by duodenal cancer (6 patients, 7.6%), cholangiocarcinoma (4 patients,5.1%) and other tumors (7 patients, 8.9%). Owing to the retrospective nature and limited patient numbers in tumors other than pancreatic cancer, it is difficult to compare the differences among different cancers. However, we found that there is no statistic difference in survival time [121 (11-713) vs. 78 (10-887) days, p = 0.737] and stent patency time [116 (10-671) vs. 70 (7-564), p = 0.854] between pancreatic cancer and non-pancreatic cancers in our study. We have added the information in the manuscript. Please see page 14, line 235-238.

3.The authors only mentions the length of the stenosis, while no the degree of stenosis. The degree of stenosis in the same part of the stenosis is different, and the effect after placement is also different.

Reply: Thank you for the comments. In our study, lesions that were considered total obstructed when a standard upper endoscope cannot pass through the obstruction site. Only patients with total obstruction of duodenum on endoscopic evaluation were enrolled for stent placement. Therefore, only the length of stenosis was measured under fluoroscopy in our study. 

4.The type of stent placed was not specified. The same type of stent bears the same support force and different stents bear different support forces.

Reply: Thank you for the comments. In our study cohort, the WallFlex Single-use Duodenal Stent (Boston Scientific Corporation, USA) or Bonastent (Standard SCI. Tech Inc, South Korea) were used (21 vs. 56 patients, respectively). There was no significant difference in terms of survival time [120.0 (11-541) vs. 118.5 (10-887), p = 0.455] and stent patency time [106.0 (11-541) vs. 106.0 (7-671), p = 0.544] between the two stent types. The information is added in the text. Please see page 15, line 238-242.

Reviewer #2: The work Outcomes of Patients with Malignant Duodenal Obstruction After Receiving Self-Expandable Metallic Stents: A Single Center Experience summarized data from 79 malignant duodenal obstruction patients treated with self-Expandable Metallic Stents. Study is properly designed and executed, and I have few comments for the work.

1.First of all, author used ECOG score to indicate patients’ health condition. Table 1 and 3 should follow Table S1 change ECOG to ECOG score. Or the abbreviation of the Oncology Group will not give any meaning for the numbers.

Reply: Thank you for your kindest suggestion. We have adjusted the term “ECOG” to “ECOG scores” in revised Table 1 and 3 accordingly.

2.Second, GOOSS score was summarized till day 30. Since patients did not receive chemotherapy after put in stent has a median survival around 60 days. Is there any GOOSS score data for patients till day 60, 90?

Reply: Thank you for your comments. In our study, the GOOSS score was recorded only till day 30. After one month, we did not routinely follow up the condition of oral intakes and only patients who had symptoms of restenosis would receive re-evaluation. Therefore, most of the data at day 60 and 90 are missing. Thanks again for the useful suggestions. We will keep following up the GOOSS scores at day 60 and day 90 for patients who receive stent insertion and perform further analysis in future studies. 

3.Last but not the least, GOOSS score largely influenced patient recovery due to the nutrient acquired always determined by diet. To understand the recovery progress, is it possible to include a GOOSS score chart with detailed classification.

Reply: Thank you for the suggestion. The detailed classifications are added in the S1 Table in the supplementary data.

---

## [Decision Letter · Decision Letter 1]

11 May 2022

Outcomes of Patients with Malignant Duodenal Obstruction After Receiving Self-Expandable Metallic Stents: A Single Center Experience

PONE-D-22-02289R1

Dear Dr. Kuei-Chuan Lee,

We’re pleased to inform you that your manuscript has been judged scientifically suitable for publication and will be formally accepted for publication once it meets all outstanding technical requirements.

Kind regards,

Wenguo Cui, Ph.D

Academic Editor

PLOS ONE

Additional Editor Comments (optional):

Reviewers' comments:

Reviewer's Responses to Questions

**Comments to the Author**

1. If the authors have adequately addressed your comments raised in a previous round of review and you feel that this manuscript is now acceptable for publication, you may indicate that here to bypass the “Comments to the Author” section, enter your conflict of interest statement in the “Confidential to Editor” section, and submit your "Accept" recommendation.

Reviewer #1: All comments have been addressed

Reviewer #2: All comments have been addressed

2. Is the manuscript technically sound, and do the data support the conclusions?

Reviewer #1: Yes

Reviewer #2: Yes

3. Has the statistical analysis been performed appropriately and rigorously? 

Reviewer #1: Yes

Reviewer #2: Yes

4. Have the authors made all data underlying the findings in their manuscript fully available?

Reviewer #1: Yes

Reviewer #2: Yes

5. Is the manuscript presented in an intelligible fashion and written in standard English?

Reviewer #1: Yes

Reviewer #2: Yes

6. Review Comments to the Author

Reviewer #1: The revised manuscript is able to clearly answer the relevant questions. Moreover, please also explain the statistical method in the text clearly. Other than that, no other problem.

Reviewer #2: Author addressed all the comments from both reviewers in current revision. This center summarized data from 79 Malignant Duodenal Obstruction patients across almost 12 years. This summary could provide insightful information for future MDO patient treatment.

7. PLOS authors have the option to publish the peer review history of their article (what does this mean?). If published, this will include your full peer review and any attached files.

Reviewer #1: No

Reviewer #2: No

---

## [Editor Report · Acceptance letter]

17 May 2022

PONE-D-22-02289R1 

Outcomes of Patients with Malignant Duodenal Obstruction After Receiving Self-Expandable Metallic Stents: A Single Center Experience 

Dear Dr. Lee:

I'm pleased to inform you that your manuscript has been deemed suitable for publication in PLOS ONE. Congratulations! Your manuscript is now with our production department. 

Kind regards, 

on behalf of

Professor Wenguo Cui 

Academic Editor

PLOS ONE